# Anticancer and Antiphytopathogenic Activity of Fluorinated Isatins and Their Water-Soluble Hydrazone Derivatives

**DOI:** 10.3390/ijms242015119

**Published:** 2023-10-12

**Authors:** Andrei V. Bogdanov, Margarita Neganova, Alexandra Voloshina, Anna Lyubina, Syumbelya Amerhanova, Igor A. Litvinov, Olga Tsivileva, Nurgali Akylbekov, Rakhmetulla Zhapparbergenov, Zulfiia Valiullina, Alexandr V. Samorodov, Igor Alabugin

**Affiliations:** 1Arbuzov Institute of Organic and Physical Chemistry, FRC Kazan Scientific Center, Russian Academy of Sciences, Akad. Arbuzov St. 8, Kazan 420088, Russia; neganovam@ipac.ac.ru (M.N.); microbi@iopc.ru (A.V.); aplyubina@gmail.com (A.L.); syumbelya07@mail.ru (S.A.); litvinov@iopc.ru (I.A.L.); ialabugin@gmail.com (I.A.); 2Institute of Physiologically Active Compounds at Federal Research Center of Problems of Chemical Physics and Medicinal Chemistry, Russian Academy of Sciences, Severnij Pr. 1, Chernogolovka 142432, Russia; 3Institute of Biochemistry and Physiology of Plants and Microorganisms, Saratov Scientific Centre of the Russian Academy of Sciences, Entuziastov Ave. 13, Saratov 410049, Russia; tsivileva_o@ibppm.ru; 4Laboratory of Engineering Profile “Physical and Chemical Methods of Analysis”, Korkyt Ata Kyzylorda University, Aitekebie Str. 29A, Kyzylorda 120014, Kazakhstan; nurgali_089@mail.ru; 5Department of Pharmacology, Bashkir State Medical University, Lenin St. 8, Ufa 450008, Russia; z_suleimanova@mail.ru (Z.V.); avsamorodov@gmail.com (A.V.S.); 6Department of Chemistry and Biochemistry, Florida State University, 95 Chieftan Way, Tallahassee, FL 32306-4390, USA

**Keywords:** isatin, quaternary ammonium compounds, cancer, hydrazones, crystal structure, antiphytopathogenes, cytotoxicity

## Abstract

A series of new fluorinated 1-benzylisatins was synthesized in high yields via a simple one-pot procedure in order to explore the possible effect of ortho-fluoro (**3a**), chloro (**3b**), or bis-fluoro (**3d**) substitution on the biological activity of this pharmacophore. Furthermore, the new isatins could be converted into water-soluble isatin-3-hydrazones using their acid-catalyzed reaction with Girard’s reagent P and its dimethyl analog. The cytotoxic action of these substances is associated with the induction of apoptosis caused by mitochondrial membrane dissipation and stimulated reactive oxygen species production in tumor cells. In addition, compounds **3a** and **3b** exhibit platelet antiaggregation activity at the level of acetylsalicylic acid, and the whole series of fluorine-containing isatins does not adversely affect the hemostasis system as a whole. Among the new water-soluble pyridinium isatin-3-acylhydrazones, compounds **7c** and **5c**,**e** exhibit the highest antagonistic effect against phytopathogens of bacterial and fungal origin and can be considered useful leads for combating plant diseases.

## 1. Introduction

The isatin heterocyclic system is the precursor of a large number of derivatives [1,2] with a wide range of biological and pharmacological properties [3,4,5,6]. A number of properties reported for this class of compounds include anticonvulsant [7,8], antistress and anxiogenic [9], antiviral [10], antimicrobial [11], antituberculous [12], antimalarial [13], antifungal [14], and antibacterial [15], which determine their main areas of use. Among the many pharmacological or medical applications of isatin and its derivatives (Figure 1), their use as antitumor agents deserves special attention. Several isatin derivatives have already passed clinical trials and have become approved anticancer drugs [16]. In particular, the most promising isatin derivative, sunitinib, has been clinically approved by the FDA for the treatment of gastrointestinal stromal tumors, renal cell carcinoma, and a rare type of pancreatic cancer [17]. However, efforts are continuing to diversify chemical modifications and achieve deeper insights into the mechanism of biological action of isatin derivatives. Further rational research is needed to create more effective drugs with less systemic toxicity, better pharmacological characteristics, and a clear mechanism of action.

One of the limiting factors in the medical and biological applications of many isatin derivatives is their low water solubility. One approach to solving this problem is based on the introduction of an ammonium moiety. Not only does this positively charged group improve the desired physicochemical characteristics, but it can also expand the range of action of the compounds. This is evidenced by recent reviews on the therapeutic possibilities of various quaternary ammonium compounds [18,19,20,21,22]. Earlier, we also demonstrated the high potential of isatin hydrazones containing a quaternary ammonium center in the search for drug candidates for combating human and plant diseases [23,24,25,26,27] (Figure 2). 

Another advantage of isatins as a promising platform for the design and development of new drugs is the possibility of introducing additional substituents at positions of the isatin ring. One of the popular approaches to chemical modifications is the introduction of halogen atoms. For example, approximately 25–30% of new drugs on the market contain at least one fluorine atom [28]. This modification often leads to a significant increase in an existing type of biological activity and/or an expansion of the action spectrum [29].

In numerous systematic reviews and meta-analyses, there is a large amount of evidence that excessive platelet count contributes to the progression of malignant neoplasms [30,31,32]. This is due to the formation of specific interactions between platelets and neoplastic cells, which leads to increased survival of tumor cells in the bloodstream, increased spread of tumor foci, and, as a result, metastasis in places remote from the primary occurrence [33]. In this regard, considerable attention of researchers is focused on the use of the well-known anticoagulants as adjunctive therapy for oncological diseases, which is an irreversible inhibitor of platelet cyclooxygenase [34], and a number of clinical studies have demonstrated its positive clinical effects in the treatment of prostate cancer [35], mammary glands [36], stomach [37], etc. That is why the presence of antiplatelet properties as a mechanism of action for potential antitumor agents is considered a promising approach to the development of therapeutic agents to combat oncopathologies.

Various modifications of isatin molecules can lead to the appearance of different desired properties. For example, phosphate derivatives of isatin hydrazones have been shown to have higher activity against sugar cane phytopathogenic fungi than some currently used synthetic fungicides [38]. Evaluation of the fungicidal activity of dialkylphosphorylhydrazones showed that several compounds are able to effectively inhibit the growth of *Rhizoctonia solani* and *Fusarium oxysporum*. In addition, these compounds did not interfere with the germination of lettuce seeds, which indicates the absence of phytotoxicity and makes them potential leaders in the search for new fungicides [39].

Complexation of several compounds with isatin was demonstrated to yield good antibacterial potential, frequently based on the synergic effects of isatin combinations against selected microorganisms [40]. The current list of bacterial pathogens isolated from various crop plants and commonly grown vegetables frequently comprises *Micrococcus luteus*, *Pseudomonas fluorescens*, *Pectobacterium carotovorum* (*Erwinia carotovora*), *Xanthomonas campestris*, and *Pectobacterium atrosepticum* (*Erwinia carotovora* subsp. *atrosepticum*) [41,42,43]. The top five cereal bacterial pathogens on the list include *Pseudomonas* and *Xanthomonas* [44].

Fungal pathogens cause 70–80% of all plant diseases [45], possessing the potential to cause large-scale disease outbreaks in a very limited period of time. Damage from these pathogens results in the destruction of the roots until the entire plant dies and in the eventual collapse of whole infected plants [46]. Therefore, the search for novel antifungal agents has been a constant hot research topic in pesticide development, especially because long-term usage of the same antifungal agent often leads to an increase in the resistance of phytopathogenic fungi [47]. 

In this work, we present novel fluorinated 1-benzylisatins and water-soluble hydrazones as a platform for the search for potential antitumor and antiphytopathogenic compounds.

## 2. Results and Discussion

### 2.1. Chemistry

#### 2.1.1. Synthesis of Diversely Fluorinated 1-benzylisatins

Fluoro-substituted derivatives **3a**–**f** were synthesized by alkylation of 5-fluoro-isatin sodium salt **2** with halogen-containing benzyl halides (Figure 1). Isatins **3a**,**c**,**e**,**f** contained only F-substituents but in different substitution patterns. In addition, we have prepared a mono-chloro isatin **3b** and a mixed 2-chloro-6-fluoro derivative **3d** in order to assess the contribution of the fluorine atom in the pendant benzene ring to the biological activity.

The halogenated isatins **3a**–**f** were isolated in high yields in their pure form immediately after the workup of the reaction mixtures. Their structure and purity have been unequivocally proven by IR and NMR spectroscopy and elemental analysis data (Appendix A).

To further assess the effect of the position of the fluorine atom and the electronic structure of substituents in the benzyl fragment on the level of antitumor activity, a small series of derivatives **4a**–**e** was obtained by analogy (Figure 2). In contrast to the reactions described in Figure 1, the alkylation of sodium salt **2** by less electrophilic benzylic halides requires heating in order to achieve higher yields of the target compounds.

In continuation of our research on the synthesis and biological activity of water-soluble isatin-3-acylhydrazones (for example, see [23,24,25,26]), we also synthesized several new pyridinium acylhydrazones (**5a**–**e**) and previously published more lipophilic analogs (**7a**–**c**) [26] (Figure 3) containing fluorine atoms in various molecular fragments (Figure 3).

Compounds **5a**–**e** and **7a**–**c** were isolated in high yields (72–90%) after spontaneous cooling of the reaction mixture. They are yellow powders that are soluble in water, aqueous DMSO, and DMF but insoluble in non-polar solvents such as ethanol, chloroform, etc.

#### 2.1.2. X-ray Study

According to X-ray data (Figure 4A,B, Appendix A, Appendix A), compounds **3f** and **4c** crystallize in different orthorhombic space groups: compound **3f** crystallizes in the non-centrosymmetric racemic space group *Pna2*_1_, and compound **4c** crystallizes in the centrosymmetric space group *Pbca*.

The conformations of the molecules in the crystals are similar: the isatin fragments are planar, and the nitrogen atom N1 has planar trigonal coordination, as is typical for amides. The benzyl substituents at the nitrogen atom are positioned non-symmetrically along the N–C8 and C8–C9 bonds. Torsion angles C2–N1–C8–C9 and N1–C8–C9–C10(C14) in the molecule are **3f** −110.3(5)^o^ and −114.0(5)^o^, while in the molecule **4c**, they are −109.5(2)° and −132.4(2)°, respectively. This arrangement places one of the hydrogen atoms at the C8 atom in a nearly eclipsed conformation with the plane of the pendant benzene ring. In this case, the main geometric parameters (bond lengths and bond angles) in molecules **3f** and **4c** are usual.

Analysis of short contacts in crystals shows multiple C–H…F intermolecular contacts, contributing to crystal packing for these compounds. Such contacts can be considered non-conventional H-bonds [48]. This observation suggests that the C–F bonds may participate in similar supramolecular interactions with their biological targets.

Thus, using simple and efficient synthetic procedures, we obtained various fluorine-containing oxindole derivatives, which can be divided into two main groups: isatins and isatin-3-hydrazones. In the next step, we proceeded to study their biological activity. 

### 2.2. Biological Studies

In this work, we approached it from two main angles: for substituted isatins, we evaluated the potential of using these substances as antitumor drugs, and for pyridinium hydrazones, we assessed their prospects as antiphytopathogenic agents (Figure 5).

#### 2.2.1. Anticancer Activity

The antitumor potential of the isatin derivatives synthesized in this work (**3a**–**6c**) and previously obtained (**8a**–**8d**) [26] (Figure 6) was primarily assessed by the ability of the compounds to reduce the survival rate of M-HeLa and HuTu 80 tumor-derived cells relative to that for normal (Chang liver) cells. Further, in order to understand the possible mechanisms of the cytotoxic action of the most promising substances, the ability to induce apoptosis and reactive oxygen species (ROS) production was determined. Additionally, the anticoagulant and antiaggregation properties were studied. 

##### Cytotoxicity of Test Compounds

The toxic effects of all test compounds on cancer and normal cells were evaluated. The cytotoxicity of synthesized isatins was compared with the currently used antitumor drug 5-fluorouracil (5-FU) [49].

All test compounds exhibit moderate activity against the cancer lines used in the experiments (Table 1, Appendix A). Compounds **3a**, **3b**, and **3d**, containing an ortho-substituted benzyl fragment with fluorine (**3a**), chlorine (**3b**), or both (**3d**) atoms, have the highest activity against all tumor cell lines. It should be noted that for substance **3b**, the cytotoxicity is 2.0 times higher than the activity of the reference drug 5-FU. Moreover, compounds **3a**, **3b**, and **3d** had low cytotoxicity against the healthy cell line WI38.

The selectivity of compounds for cancer cells is an important criterion for evaluating a potential drug. The calculated selectivity values (SI) for the most active compounds were 1.8 for M-HeLa for **3b**, 1.5 for **3d**, and HuTu-80 for **3a**—2.5, **3b**—1.5, and **3d**—2.5, for 5-FU ≤ 1. The selectivity values (SI) indicated in the table prove that the halogenated isatins exhibit better selectivity than the reference drug 5-FU.

##### Apoptosis

It is well known that tumor cells are able to avoid apoptosis, which leads to uncontrolled, abnormal proliferation and rapid tumor growth. Substances capable of inducing apoptotic death in tumor cells are promising agents in the fight against cancer [50]. Flow cytometry (Guava easyCyte, Merck, Rahway, NJ, USA) showed that the most cytotoxic compounds (**3a**, **3b**, and **3d**) were able to induce apoptosis in HuTu-80 cells. Figure 7 shows the apoptotic effects induced by the selected substances at IC_50_/2 and IC_50_ cytotoxicity concentrations. 

It can be seen from the presented data that after 24-hour incubation of HuTu 80 cells with compounds **3a**, **3b**, and **3d**, the number of cells at both the early and late stages of apoptosis increases with concentration. Molecule **3b** exhibits the highest apoptosis-inducing activity (IC_50_ concentration of 40 μM). Moreover, in the presence of these compounds, cells exist predominantly in the stage of late apoptosis.

##### Mitochondrial Membrane Potential

As can be seen in Figure 8, after 24-h incubation of HuTu 80 cells with the tested substances, a decrease in the mitochondrial membrane potential is observed, which becomes more pronounced at concentrations corresponding to the IC_50_ values of cytotoxicity. Compounds **3b** and **3d** showed the highest activity in this test, under the action of which the green fluorescence increased up to 55% relative to the control. 

Thus, these results confirm that the mechanism of action of compounds **3a**, **3b**, and **3d** is associated with the induction of apoptosis due to the ability of substances to dissipate the mitochondrial membrane.

##### Reactive Oxygen Species Production

An increase in the production of reactive oxygen species (ROS) by the halogenated isatins is another characteristic feature of the development of apoptosis along the mitochondrial pathway. Mitochondria are both a potential source and a target of ROS. ROS are known to lead to disruption of mitochondrial functions and, consequently, irreversible cell damage [51].

We have evaluated the effect of compounds **3a**, **3b**, and **3d** in HuTu 80 cells on the induction of ROS production using a flow cytometry assay and the CellROX^®^ Deep Red flow cytometry kit. The results are presented in Figure 9.

##### Anticoagulant and Antiaggregation Activity Studies

In this work, for the three most promising compounds (**3a**, **3b,** and **3d**), anticoagulant and antiaggregation properties were studied (Table 2).

Among the studied classes of compounds, derivatives **3a** and **3b** showed antiaggregation activity at the level of acetylsalicylic acid, reducing the maximum platelet aggregation by 12% on average. At the same time, compounds **3a** and **3b**, in contrast to acetylsalicylic acid, lengthen the lag-period by more than 10% relative to the control. From the point of view of anticoagulant properties, all compounds, including **3d**, showed a different level of influence on the plasma component of the hemostasis system, which manifested itself in a change in only the indicator of the internal blood coagulation pathway—activated partial thromboplastin time (APTT). Thus, fluorine-containing isatins and corresponding water-soluble acylhydrazones have high potential as scaffolds for the development of effective anticoagulant and antiaggregation agents.

The isatin derivatives synthesized in this work have the potential to be used as a basis for the design of antitumor agents with a known mechanism of action. The most cytotoxic compounds are capable of inducing the production and accumulation of reactive oxygen species by disrupting the normal functioning of mitochondria and dissipating the mitochondrial membrane. These pathological processes, provoked by the action of substances, lead to the launch of a cascade of cell death along the internal pathway of apoptosis, which is directly associated with mitochondria.

#### 2.2.2. Evaluation of Antiphytopathogenic Activity of Water-Soluble Hydrazones

##### Antibacterial Activity

We examined the synthesized isatin derivatives for their antibacterial effects against the above-tested organisms. Along with the solutions of **5a**–**e** and **7a**–**c**, test compounds were sodium hypochlorite (1000 µg/mL), chlorohexidin (500 µg/mL), and norfloxacin (500 µg/mL), a synthetic fluoroquinolone with broad-spectrum antibacterial activity against most bacteria [52]. The in vitro bactericidal activity of the target compounds **5a**–**e** and **7a**–**c** against five representative phytopathogens is summarized in Appendix A.

The results showed bactericidal activity distinct from zero against *M. luteus* B-109, *P. atrosepticum* 1043, *P. carotovorum* subsp. *carotovorum* MI, *Ps. fluorescens* EL-2.1, and *X. campestris* B-610 for all the compounds tested.

Compounds **5a**, **5b**, **5d**, and **7a** showed moderate bactericidal activity (Appendix A).

Even though four compounds (**5a**, **5b**, **5d**, and **7a**) demonstrated non-zero bactericidal action, there were worse results in **7a** when affecting *M. luteus*, and in **5b**, **5d** vs. *X. campestris*, the inhibition zone width was less than 3 mm. The higher antibacterial potential was revealed for hydrazone **7a**, with a broader spectrum against *P. carotovorum* subsp. *carotovorum* MI, *Ps. fluorescens* EL-2.1, and *X. campestris* B-610, as well as a wider inhibition zone (5–6 mm).

Compounds **5c**, **5e**, **7b**, and **7c** could be arranged into a group exhibiting a more potent ability to combat bacterial pathogens under study (Appendix A, Figure 10).

The structural peculiarity of the compounds **7a**, **7b**, and **7c** is that they are 2,3-dimethylpyridin-1-ium salts, whereas all other hydrazones (**5a**, **5b**, **5c**, **5d**, and **5e**) are pyridin-1-ium salts. However, the substances **5c**, **5e**, and **7b**, with distinctly structured pyridinium fragments, displayed virtually the same bactericidal effect assessed by the inhibition zone range of 4 to 7 mm, dependent on the test system used (Appendix A).

Greater antibacterial activity by the given hydrazone seems to be connected to the presence of the trifluoromethyl group on the phenyl ring in the side-chain substituent. Trifluoromethylated compounds are of utmost interest due to their unique properties, including their in vitro antifungal activity [53]. However, a few previous works reported that the complicated structured trifluoro derivatives of indole displayed minimal antifungal activity against *Fusaruim*, with an inhibition rate of about 73% at an as high concentration as 500 mcg/mL [54]. In our work, it was compound **7c** containing a 3-trifluoromethylbenzyl substituent at the one position of a 2-oxoindolin moiety that showed the best-in-experiment bactericidal activity. Other hydrazones from this group, **5c**, **5e**, and **7b**, displayed a slightly diminished bactericidal effect and formed a 5-mm to 7-mm zone of the pathogens’ growth inhibition (Appendix A). 

Taking into account the data on the values of the inhibition zone width (mm), it is conditionally possible to arrange the studied 2 mM solutions of compounds **5a**–**e**, **7a**–**c** in descending order of activity as follows: **7c** (6–9 mm) > **5e** (4–7 mm) ≈ **5c** (4–7 mm) ≈ **7b** (4–7 mm) > **5a** (3–6 mm) > **7a** (2–6 mm) > **5b** (2,5–4 mm) > **5d** (2–4 mm). Thus, with respect to the phytopathogens used, the most profound activity was shown by hydrazone **7c**.

##### Antifungal Activity

In this study, a series of fluorine-containing pyridinium isatin hydrazones (**5a**–**e** and **7a**–**c**) was prepared, and their antifungal activities were evaluated. Along with the solutions of hydrazones, the test compounds were fludioxonil and N-cetylpyridinium chloride. These commercial, widely used fungicides capable of inhibiting the pathogenic fungi mycelium propagation served for the purpose of comparison in the course of the fungicidal effect assays. Fludioxonil is one of the most potent fungicides against the diseases of agricultural crops caused by the representatives of both *Fusarium* spp. and *Phytophthora* spp. [55]. Under laboratory conditions, the assessment of the phytopathogenic fungi’s resistance to fludioxonil is conducted at concentrations ranging from 0.1 to 10 μg/mL [56]. N-cetylpyridinium chloride (CPC) is a quaternary ammonium compound with broad-spectrum antimicrobial activity [57] used in topical formulations with antifungal activity [58,59].

The in vitro antifungal activity of target compounds **5a**–**e** and **7a**–**c** against the fungal pathogen *F. oxysporum* is summarized in Appendix A.

Within the entire experiment with the phytopathogen *F. oxysporum*, the highest fungicidal activity estimated through the EC_50_ indices (µg/mL) was displayed by the studied compounds **5e** (0.60) and **7a** (7.25). Antifungal activity of the rest of the hydrazones tested was also considerable, with the EC_50_ value not exceeding 21 µg/mL for **7b**. The methyl group at the C5 position of the oxindole moiety, in combination with the trifluoromethyl group on the benzyl ring (**5e**) of the 1-benzylisatin hydrazones, as well as the fluorine atom at the C2 position on this benzyl ring (**7a**), had a significant effect on the pyridinium isatin hydrazones antifungal activity against *F. oxysporum* (Figure 11).

Compound **5e** had the most significant antifungal activity against *F. oxysporum* (IC_50_ = 0.60 µg/mL).

The in vitro antifungal activity of target compounds **5a**–**e** and **7a**–**c** against the fungal pathogen *P. cactorum* is summarized in Appendix A.

Within the entire experiment with the phytopathogen *P. cactorum*, the highest fungicidal activity assessed through the EC_50_ parameter (µg/mL) was displayed by the studied compounds **5a** (15.12), **5c** (15.40), and **5b** (18.19). Antifungal activity of the rest of the tested hydrazones was also considerable, with the EC_50_ value not exceeding 28 µg/mL for **7b**. The fluorine atom (**5a**, **5b**, and **5c**) at the C5 position of the isatin skeleton in the pyridinium isatin hydrazones, in combination with the strong electron-withdrawing units: the trifluoromethyl group (-CF_3_) (**5c**), the fluorine atom at the C2 position (**5a**), or the 2-Cl-6-F (**5b**) substituent on the benzyl ring of the 1-benzylisatin hydrazones, had a significant effect on the improvement of the antimicrobial activity against *P. cactorum*. Compounds **5a** and **5c** exhibited essentially the same antifungal activity, equal to 15 μg/mL.

All of the fluorine-containing compounds (**5a**–**e** and **7a**–**c**) showed an antagonistic effect against the oomycete *P. cactorum*. This effect is somewhat superior to that of N-cetylpyridinium chloride by 1.3 (**5e**, **7b**) to 2.3 (**5a**, **5c**) times and of fludioxonil by 1.1 (**5e**, **7b**) to 2.0 (**5a**) times in 5 days of *Phytophthora* growth. At the same culture age as ascomycete *F. oxysporum*, all of the studied fluorine-containing compounds outperformed the reference fungicides in the antifungal effect of this test system. Fludioxonil was less efficient than hydrazones in inhibiting *Fusarium* growth. Inhibition by hydrazon ranged from 139 times greater than **5e** down to 4.1 times greater than **7b**. N-cetylpyridinium chloride showed weaker fungicidal action vs. *F. oxysporum*, as compared to **5a**–**e** and **7a**–**c**, by 235 times more than **5e** and down to 7.0 times more than **7b**.

When comparing the fungicidal activity of hydrazones in the *F. oxysporum* test system and their antagonistic action with respect to *P. cactorum*, one can recognize that, without exceptions, all compounds under study show the greater ability to inhibit the mycelial propagation of *F. oxysporum* (Table 3). 

The inhibiting action of fluorine-containing hydrazones **5a**–**e**, **7a**–**c** judged through the EC_50_ criterion occurred to be stronger with respect to *Fusarium* mycelial development, by 46 times more than **5e**, down to 1.1 times more than **5c**, in comparison to the same compounds’ impact on *Phytophthora*. In contrast, the reference fungicides exhibited the opposite tendency for the fungal test system used. Fludioxonil and N-cetylpyridinium chloride showed a much greater effect of *P. cactorum* growth inhibition compared to *F. oxysporum* (2.8 and 4.1 times, respectively, for the two fungicides in Table 3). The behavior of the fluorine-free compounds synthesized and examined for antiphytopathogenic activity by us earlier [25] was in line with the latter tendency, i.e., they were similar to the reference fungicides. The results could testify to a promising synthetic strategy potentially resulting in the fluorine-containing hydrazones being uncommonly efficient vs. *Fusarium oxysporum* in comparison with known fungicides. 

## 3. Materials and Methods

### 3.1. Chemistry

IR spectra were recorded on the IR Fourier spectrometer Tensor 37 (Bruker Optik GmbH, Ettlingen, Germany) in the 400–3600 cm^−1^ range in KBr. The ^1^H-, ^19^F-, and ^13^C-NMR spectra were recorded on a Bruker AVANCE 400 spectrometer (Bruker BioSpin, Rheinstetten, Germany) operating at 400 MHz (for ^1^H NMR), 377 MHz (for ^19^F NMR) and 101 MHz (for ^13^C NMR), Brucker spectrometers AVANCE*III*-500 (Bruker BioSpin, Rheinstetten, Germany) operating at 500 MHz (for ^1^H NMR) and 126 MHz (for ^13^C MMR), and Bruker Avance 600 spectrometer (Bruker BioSpin, Rheinstetten, Germany) operating at 600 MHz (for ^1^H NMR) and 151 MHz (for ^13^C NMR). Chemical shifts were measured in δ (ppm) with reference to the solvent (δ = 7.27 ppm and 77.00 ppm for CDCl_3_, δ = 2.50 ppm and 39.50 ppm for DMSO-*d*_6_, for ^1^H and ^13^C NMR, respectively). Elemental analysis was performed on a CHNS-O Elemental Analyser EuroEA3028-HT-OM (EuroVector S.p.A., Milan, Italy). The melting points were determined on the Stuart SMP10 apparatus (Stuart, Birmingham, UK).

X-ray crystallography data. Data for **3f** and **4c** were collected on a Bruker D8 QUEST with a PHOTON II CCD diffractometer (Bruker AXS, Karlsruhe, Germany), using graphite monochromated MoKα (λ = 0.71073 Å) radiation and ω-scan rotation. Dataset collection images were indexed, integrated, and scaled using the APEX3 [60] dataset reduction package and corrected for absorption using SADABS [61]. The structure was solved by direct methods and refined using the SHELX program [62]. All non-hydrogen atoms were refined anisotropically. H atoms were calculated on idealized positions and refined as riding atoms. Crystal Data and Refinement Details are presented in Appendix A (see Appendix A). The X-ray analysis was performed on the equipment of the Spectral–Analytical Center of FRC Kazan Scientific Center of RAS.

CCDC 2268269 and 2268273 (**3f** and **4c**) contain the supplementary crystallographic data for this paper. These data can be obtained free of charge via www.ccdc.cam.ac.uk/conts/retrieving.html (accessed on 16 September 2023) or from the Cambridge Crystallographic Data Centre, 12 Union Road, Cambridge CB2 1EZ, UK; fax: (+44) 1223-336-033; or deposit@ccdc.cam.uk.

Synthesis of fluorine-containing indolin-2,3-diones (isatins) **3a**–**f** and **4a**–**e** (general method). A magnetically stirred solution of 5-substituted isatin **1** (10 mmol) in dry DMF (20 mL) and NaH (10 mmol, 60% suspension in mineral oil) was added in portions for 30 min at 5 °C (ice-water external bath). During the addition of sodium hydride, the reaction mixture turns purple. After 30 min, the corresponding benzyl halide (10 mmol) was added dropwise, followed by additional stirring at r.t. (3 h for **3a**–**f** and 6 h for **4a**–**e**). Then, a solution was poured into an ice/water mixture (*w*/*w* 100 g/200 g), and the precipitate that formed was filtered off, washed consecutively with cold water (50 mL), light petroleum (40 mL), and dried in vacuo (12 Torr). For better precipitation of the target product in the case of alkylbenzyl isatins, sodium chloride (10 g) was added to the mixture of ice and water.

5-Fluoro-1-(2-fluorobenzyl)indoline-2,3-dione (**3a**). Orange powder. Yield 95%, m.p. = 120–121 °C. IR spectrum, ν, cm^−1^: 1333 (C–F), 1621 (C=C), 1732 (C=O), 3066 (CH). ^1^H NMR (600 MHz, CDCl_3_) δ 7.37–7.28 (m, 3H, Ph), 7.26–7.21 (m, 1H, Ar), 7.15–7.08 (m, 2H, Ar), 6.88 (dd, *J* = 8.6 Hz, *J* = 3.6 Hz, 1H, Ar), 4.97 (s, 2H, CH_2_). ^13^C NMR (151 MHz, CDCl_3_) δ 182.4, 160.8 (d, *J* = 176.9 Hz, C–F), 159.2 (d, *J* = 176.5 Hz, C–F), 158.1, 146.5, 130.3 (d, *J* = 8.1 Hz, C–F, CH), 130.0 (d, *J* = 1.8 Hz, C–F, CH), 124.9 (d, *J* = 2.4 Hz, C–F, CH), 124.8 (d, *J* = 24.2 Hz, C–F, CH), 121.4 (d, *J* = 14.3 Hz, C–F), 118.3 (d, *J* = 6.9 Hz, C–F), 115.8 (d, *J* = 21.6 Hz, C–F, CH), 112.4 (d, *J* = 24.3 Hz, C–F, CH), 111.9 (d, *J* = 3.1 Hz, C–F, CH), 37.4 (d, *J* = 4.0 Hz, C–F, CH_2_)). ^19^F NMR (600 MHz, DMSO-*d*_6_) δ −119.9, −119.5. Found: C, 65.87; H, 3.34; F, 13.83; N, 5.11. Anal. calcd (%) for C_15_H_9_F_2_NO_2_: C, 65.94; H, 3.32; F, 13.91; N, 5.13.

1-(2-Chlorobenzyl)-5-fluoroindoline-2,3-dione (**3b**). Light orange powder. Yield 76%, m.p. = 167–168 °C. IR spectrum, ν, cm^−1^: 746 (CCl), 1344 (C–F), 1620 (C=C), 1733 (C=O), 3061 (CH). ^1^H NMR (500 MHz, CDCl_3_) δ 7.43 (d, 1H, *J* = 7.4 Hz, Ar), 7.34 (dd, *J* = 6.5 Hz, *J* = 2.7 Hz, 1H, Ar), 7.29–7.19 (m, 4H, Ar), 6.74 (dd, *J* = 8.6 Hz, *J* = 3.5 Hz, 1H, Ar), 5.06 (s, 2H, CH_2_). ^13^C NMR (126 MHz, DMSO-*d*_6_) δ 182.1 (d, *J* = 1.6 Hz, C–F), 158.6 (d, *J* = 241.2 Hz, C–F), 158.5, 146.4, 132.2 (CH), 131.8, 129.4 (d, *J* = 22.8 Hz, C–F, CH), 128.3 (CH), 127.4 (CH), 123.9, 123.7, 118.9 (d, *J* = 7.0 Hz, C–F), 112.2 (d, *J* = 7.6 Hz, C–F, CH), 111.5 (d, *J* = 24.4 Hz, C–F, CH), 41.2 (CH_2_). ^19^F NMR (600 MHz, DMSO-*d*_6_) δ −119.8. Found: C, 62.21; H, 3.15; Cl, 12.19; F, 6.46; N, 4.78. Anal. calcd (%) for C_15_H_9_ClFNO_2_: C, 62.19; H, 3.13; Cl, 12.24; F, 6.56; N, 4.84.

1-(2,6-Difluorobenzyl)-5-fluoroindoline-2,3-dione (**3c**). Red powder. Yield 85%, m.p. 131–132 °C. IR spectrum, ν, cm^−1^: 1327 (C–F), 1623 (C=C), 1735 (C=O), 1753 (C=O), 3057 (CH). ^1^H NMR (500 MHz, CDCl_3_) δ 7.34–7.28 (m, 2H, Ar), 7.24 (ddd, *J* = 8.8 Hz, *J* = 8.7 Hz, *J* = 2.7 Hz, 1H, Ar), 6.94 (dd, *J* = 8.1 Hz, *J* = 8.0 Hz, 1H, Ar), 6.88 (dd, *J* = 8.7 Hz, *J* = 3.6 Hz, 1H, Ar), 5.02 (s, 2H, CH_2_). ^13^C NMR (126 MHz, CDCl_3_) δ 182.3, 161.5 (dd, ^1^*J* = 250.4 Hz, ^3^*J* = 7.3 Hz, C–F), 159.3 (d, *J* = 246.1 Hz, C–F), 157.3 (d, *J* = 0.6 Hz, C–F), 146.3 (d, *J* = 2.0 Hz), 130.7 (t, *J* = 10.4 Hz, C–F, CH), 124.7 (d, *J* = 24.2 Hz, C–F, CH), 118.4 (d, *J* = 7.0 Hz, C–F), 112.4 (dd, *J* = 24.2 Hz, *J* = 0.3 Hz, C–F, CH), 111.8 (dd, *J* = 25.1 Hz, *J* = 6.1 Hz, C–F, CH), 111.5 (dt, *J* = 9.2 Hz, *J* = 2.4 Hz, C–F, CH), 109.9 (t, *J* = 18.2 Hz, C–F), 32.3 (t, *J* = 4.0 Hz, C–F, CH_2_). Found: C, 61.80; H, 2.80; F, 19.45; N, 4.77. Anal. calcd (%) for C_15_H_8_F_3_NO_2_: C, 61.86; H, 2.77; F, 19.57; N, 4.81.

1-(2-Chloro-6-fluorobenzyl)-5-fluoroindoline-2,3-dione (**3d**). Bright orange powder. Yield 90%, m.p. = 140–141 °C. IR spectrum, ν, cm^−1^: 779 (CCl), 1330 (C–F), 1624 (C=C), 1745 (C=O), 1772 (C=O), 3070 (CH). ^1^H NMR (500 MHz, CDCl_3_) δ 7.32–7.27 (m, 3H, Ar), 7.20 (ddd, *J* = 8.7 Hz, *J* = 8.7 Hz, *J* = 2.7 Hz, 1H, Ar), 7.07–7.02 (m, 1H, Ar), 6.80 (dd, *J* = 8.7 Hz, *J* = 3.6 Hz, 1H, Ar), 5.12 (d, *J* = 0.8 Hz, 2H, CH_2_). ^13^C NMR (126 MHz, DMSO-*d*_6_) δ 182.2 (d, *J* = 0.7 Hz, C–F), 161.4 (d, *J* = 249.7 Hz, C–F), 158.4 (d, *J* = 241.6 Hz, C–F), 157.8, 146.8, 134.3 (d, *J* = 5.3 Hz, C–F, CH), 130.9 (d, *J* = 10.0 Hz, C–F), 125.9 (d, *J* = 2.9 Hz, C–F, CH), 124.3 (d, *J* = 24.2 Hz, C–F, CH), 120.3 (d, *J* = 16.4 Hz, C–F), 118.5 (d, *J* = 7.4 Hz, C–F), 114.9 (d, *J* = 22.5 Hz, C–F, CH), 112.0 (d, *J* = 6.3 Hz, C–F, CH), 111.6 (d, *J* = 24.4 Hz, C–F, CH), 36.2 (d, *J* = 3.2 Hz, C–F, CH_2_). ^19^F NMR (600 MHz, DMSO-*d*_6_) δ −119.8, −111.0. Found: C, 58.60; H, 2.59; Cl, 11.48; F, 12.23; N, 4.54. Anal. calcd (%) for C_15_H_8_ClF_2_NO_2_: C, 58.56; H, 2.62; Cl, 11.52; F, 12.35; N, 4.55.

5-Fluoro-1-(3-(trifluoromethyl)benzyl)indoline-2,3-dione (**3e**). Orange powder. Yield 83%, m.p. = 170–171 °C. IR spectrum, ν, cm^−1^: 1330 (C–F), 1621 (C=C), 1731 (C=O), 1745 (C=O), 3066 (CH). ^1^H NMR (400 MHz, DMSO-*d*_6_) δ 7.85 (s, 1H, Ar), 7.75 (d, *J* = 7.7 Hz, 1H, Ar), 7.65 (d, *J* = 7.8 Hz, 1H, Ar), 7.58 (t, *J* = 7.7 Hz, 1H, Ar), 7.49 (dd, *J* = 7.2 Hz, *J* = 2.7 Hz, 1H, Ar), 7.45 (ddd, *J* = 9.3 Hz, *J* = 8.8 Hz, *J* = 2.8 Hz, 1H, Ar), 6.98 (dd, *J* = 8.6 Hz, *J* = 3.8 Hz, 1H, Ar), 5.01 (s, 2H, CH_2_). ^13^C NMR (101 MHz, DMSO-*d*_6_) δ 182.2, 158.5 (d, *J* = 241.3 Hz, C–F), 158.6, 146.2, 137.0, 131.4 (CH), 129.7 (CH), 129.3 (q, *J* = 31.6 Hz, C–F), 124.3 (m, CH), 124.1 (m, CH), 123.6 (d, *J* = 24.1 Hz, C–F, CH), 118.9 (d, *J* = 7.3 Hz, C–F), 112.1 (d, *J* = 7.6 Hz, C–F, CH), 111.5 (d, *J* = 22.4 Hz, C–F, CH), 42.5 (CH_2_). Found: C, 59.40; H, 2.83; F, 23.47; N, 4.30. Anal. calcd (%) for C_16_H_9_F_4_NO_2_: C, 59.45; H, 2.81; F, 23.51; N, 4.33.

5-Fluoro-1-((perfluorophenyl)methyl)indoline-2,3-dione (**3f**). Purple powder. Yield 75%, m.p. = 112–113 °C. IR spectrum, ν, cm^−1^: 1330 (C–F), 1625 (C=C), 1749 (C=O), 3054 (CH). ^1^H NMR (500 MHz, CDCl_3_) δ 7.37–7.30 (m, 2H, Ar), 6.89 (dd, *J* = 8.6 Hz, *J* = 3.4 Hz, 1H, Ar), 5.04 (s, 2H, CH_2_). ^13^C NMR (126 MHz, CDCl_3_) δ 181.5 (d, *J* = 1.8 Hz, C–F), 159.5 (d, *J* = 247.1 Hz, C–F), 157.3, 151.9–149.3 (m, C–F), 146.8–144.3 (m, C–F), 145.6 (d, *J* = 1.7 Hz, C–F), 142.9–140.4 (m, C–F), 136.8–136.2 (m, C–F), 124.9 (d, *J* = 24.3 Hz, C–F, CH), 118.5 (d, *J* = 6.9 Hz, C–F), 112.8 (d, *J* = 24.3 Hz, C–F, CH), 111.1 (d, *J* = 7.1 Hz, C–F, CH), 32.1 (CH_2_). Found: C, 52.13; H, 1.56; F, 32.93; N, 4.00. Anal. calcd (%) for C_15_H_5_F_6_NO_2_: C, 52.19; H, 1.46; F, 33.02; N, 4.06.

5-Fluoro-1-(4-methylbenzyl)indoline-2,3-dione (**4a**). Orange powder. Yield 73%, m.p. = 111–112 °C. IR spectrum, ν, cm^−1^: 1330 (C–F), 1623 (C=C), 1730 (C=O), 1741 (C=O), 2923 (CH). ^1^H NMR (500 MHz, DMSO-*d*_6_) δ 7.48–7.41 (m, 2H, Ar), 7.31 (d, *J* = 8.0 Hz, 1H, Ar), 7.14 (d, *J* = 7.9 Hz, 1H, Ar), 6.95 (dd, *J* = 8.6 Hz, *J* = 3.8 Hz, 1H, Ar), 4.86 (s, 2H, CH_2_), 2.27 (s, 3H, CH_3_). ^13^C NMR (126 MHz, DMSO-*d*_6_) δ 182.5 (d, *J* = 2.0 Hz, C–F), 158.5 (d, *J* = 241.3 Hz, C–F), 158.3, 146.5 (d, *J* = 1.4 Hz, C–F), 136.7, 132.2, 129.2 (CH), 127.3 (CH), 123.8 (d, *J* = 24.2 Hz, C–F, CH), 118.6 (d, *J* = 7.3 Hz, C–F), 112.4 (d, *J* = 7.4 Hz, C–F, CH), 111.4 (d, *J* = 24.4 Hz, C–F, CH), 42.7 (CH_2_), 20.6 (CH_3_). Found: C, 71.29; H, 4.43; F, 7.00; N, 5.16. Anal. calcd (%) for C_16_H_12_FNO_2_: C, 71.37; H, 4.49; F, 7.06; N, 5.20.

1-(3,5-Dimethylbenzyl)-5-fluoroindoline-2,3-dione (**4b**). Light orange powder. Yield 70%, m.p. = 97–98 °C. IR spectrum, ν, cm^−1^: 1341 (C–F), 1623 (C=C), 1741 (C=O), 2922 (CH). ^1^H NMR (400 MHz, CDCl_3_) δ 7.30 (dd, *J* = 6.5 Hz, *J* = 2.7 Hz, 1H, Ar), 7.20 (ddd, *J* = 8.6 Hz, *J* = 8.6 Hz, *J* = 2.7 Hz, 1H, Ar), 6.93 (s, 1H, Ar), 6.91 (s, 2H, Ar), 6.75 (dd, *J* = 8.6 Hz, *J* = 3.6 Hz, 1H, Ar), 4.84 (s, 2H, CH_2_), 2.28 (s, 6H, CH_3_). ^13^C NMR (101 MHz, CDCl_3_) δ 182.8 (d, *J* = 1.7 Hz, C–F), 159.3 (d, *J* = 246.0 Hz, C–F), 158.1, 146.9 (d, *J* = 1.7 Hz, C–F), 138.8, 134.0, 129.9 (CH), 125.1 (CH), 124.6 (d, *J* = 24.1 Hz, C–F, CH), 118.3 (d, *J* = 7.0 Hz, C–F), 112.4 (d, *J* = 15.3 Hz, C–F, CH), 112.2 (d, *J* = 0.9 Hz, C–F, CH), 44.1 (CH_2_), 21.2 (CH_3_). Found: C, 72.00; H, 4.95; F, 6.60; N, 4.90. Anal. calcd (%) for C_17_H_14_FNO_2_: C, 72.07; H, 4.98; F, 6.71; N, 4.94.

1-(4-(*tert*-Butyl)benzyl)-5-fluoroindoline-2,3-dione (**4c**). Orange powder. Yield 71%, m.p. = 123–124 °C. IR spectrum, ν, cm^−1^: 1333 (C–F), 1623 (C=C), 1734 (C=O), 2959 (CH), 3063 (CH). ^1^H NMR (400 MHz, CDCl_3_) δ 7.36 (d, *J* = 8.6 Hz, 2H, Ar), 7.29 (dd, *J* = 6.5 Hz, *J* = 2.7 Hz, 1H, Ar), 7.25 (d, *J* = 8.6 Hz, 2H, Ar), 7.20 (ddd, *J* = 8.7 Hz, *J* = 8.7 Hz, *J* = 2.7 Hz, 1H, Ar), 6.78 (dd, *J* = 8.6 Hz, *J* = 3.6 Hz, 1H, Ar), 4.89 (s, 2H, CH_2_), 1.29 (s, 9H, CH_3_). ^13^C NMR (101 MHz, CDCl_3_) δ 182.7 (d, *J* = 1.9 Hz, C–F), 159.3 (d, *J* = 245.9 Hz, C–F), 158.0, 151.4, 146.9 (d, *J* = 1.6 Hz, C–F), 131.1, 127.2 (CH), 126.0 (CH), 124.5 (d, *J* = 24.1 Hz, C–F, CH), 118.2 (d, *J* = 7.0 Hz, C–F), 112.4 (d, *J* = 12.6 Hz, C–F, CH), 112.2 (d, *J* = 4.4 Hz, C–F, CH), 43.8 (CH_2_), 34.5, 31.2 (CH_3_). Found: C, 73.21; H, 5.80; F, 6.01; N, 4.43. Anal. calcd (%) for C_19_H_18_FNO_2_: C, 73.30; H, 5.83; F, 6.10; N, 4.50.

1-(3,5-Di-*tert*-butylbenzyl)-5-fluoroindoline-2,3-dione (**4d**). Orange powder. Yield 72%, m.p. = 136–137 °C. IR spectrum, ν, cm^−1^: 1342 (C–F), 1622 (C=C), 1734 (C=O), 2964 (CH), 3043 (CH). ^1^H NMR (400 MHz, CDCl_3_) δ 7.37 (s, 1H, Ar), 7.32 (dd, *J* = 6.4 Hz, *J* = 2.5 Hz, 1H, Ar), 7.21 (ddd, *J* = 8.6 Hz, *J* = 8.6 Hz, *J* = 2.4 Hz, 1H, Ar), 7.14 (s, 2H, Ar), 6.80 (dd, *J* = 8.6 Hz, *J* = 3.5 Hz, 1H, Ar), 4.90 (s, 2H, CH_2_), 1.29 (s, 18H, CH_3_). ^13^C NMR (101 MHz, CDCl_3_) δ 182.8 (d, *J* = 1.9 Hz, C–F), 159.3 (d, *J* = 245.9 Hz, C–F), 158.1 (d, *J* = 1.0 Hz, C–F), 151.8, 147.1 (d, *J* = 1.7 Hz, C–F), 133.4, 124.4 (d, *J* = 24.1 Hz, C–F, CH), 122.2 (CH), 121.8 (CH), 118.3 (d, *J* = 7.0 Hz, C–F), 112.3 (d, *J* = 29.4 Hz, C–F, CH), 112.2 (CH), 44.8 (CH_2_), 34.8, 31.4 (CH_3_). Found: C, 75.15; H, 7.10; F, 5.09; N, 3.72. Anal. calcd (%) for C_23_H_26_FNO_2_: C, 75.18; H, 7.13; F, 5.17; N, 3.81.

5-Methyl-1-(3-(trifluoromethyl)benzyl)indoline-2,3-dione (**4e**). Orange powder. Yield 86%, m.p. = 155–156 °C. IR spectrum, ν, cm^−1^: 1329 (C–F), 1620 (C=C), 1728 (C=O), 1742 (C=O), 2929 (CH), 3035 (CH). ^1^H NMR (500 MHz, DMSO-*d*_6_) δ 7.83 (s, 1H, Ar), 7.73 (d, *J* = 7.6 Hz, 1H, Ar), 7.64 (d, *J* = 7.8 Hz, 1H, Ar), 7.57 (dd, *J* = 7.7 Hz, *J* = 7.7 Hz, 1H, Ar), 7.40–7.38 (m, 2H, Ar), 6.88 (d, *J* = 8.7 Hz, 1H, Ar), 5.00 (s, 2H, CH_2_), 2.26 (s, 3H, CH_3_). ^13^C NMR (126 MHz, DMSO-*d*_6_) δ 183.0, 158.5, 147.9, 138.1 (CH), 137.2, 132.7 (CH), 131.3 (CH), 129.6, 129.3 (q, *J* = 31.7 Hz, C–F), 124.7 (CH), 124.2 (q, *J* = 3.8 Hz, C–F, CH), 124.1 (q, *J* = 3.7 Hz, C–F, CH), 124.0 (q, *J* = 272.4 Hz, CF_3_), 117.8, 110.6 (CH), 42.4 (CH_2_), 20.0 (CH_3_). Found: C, 63.87; H, 3.73; F, 17.80; N, 4.38. Anal. calcd (%) for C_17_H_12_F_3_NO_2_: C, 63.95; H, 3.79; F, 17.85; N, 4.39.

Synthesis of fluorine-containing isatin-3-acylhydrazones 5 (general method). To the mixture of substituted isatin (10 mmol) and 15 mL of absolute ethanol, Girard’s reagent P (10 mmol) and three drops of trifluoroacetic acid were successively added. The reaction solution was heated under reflux for 3 h. After spontaneously cooling to room temperature, the precipitate formed was filtered, washed with absolute ether, and dried in a vacuum.

1-(2-(2-(5-Fluoro-1-(2-fluorobenzyl)-2-oxoindolin-3-ylidene)hydrazinyl)-2-oxoethyl)pyridin-1-ium chloride (**5a**). Yellow powder. Yield 90%, m.p. = 180–182 °C. IR spectrum, ν, cm^−1^: 1358 (C–F), 1621 (C=C), 1686 (C=O), 3017 (CH), 3391 (NH). ^1^H NMR (500 MHz, DMSO-*d*_6_) δ 12.65 (s, 1H, NH), 9.23 (d, *J* = 5.2 Hz, 2H, Ar), 8.74 (t, *J* = 7.8 Hz, 1H, Ar), 8.28 (t, *J* = 6.9 Hz, 2H, Ar), 7.50 (d, *J* = 5.9 Hz, 1H, Ar), 7.42–7.30 (m, 3H, Ar), 7.25–7.16 (m, 2H, Ar), 7.12 (dd, *J* = 8.6 Hz, *J* = 3.9 Hz, 1H, Ar), 6.33 (s, 2H, CH_2_). 4.93 (s, 2H, CH_2_). ^13^C NMR (126 MHz, DMSO-*d*_6_) δ 167.8, 160.9 (d, *J* = 246.0 Hz, C–F), 160.4, 158.8 (d, *J* = 239.6 Hz, C–F), 158.1, 146.6 (CH), 146.5 (CH), 146.2, 139.2, 130.0 (d, *J* = 7.9 Hz, C–F, CH), 129.8 (d, *J* = 3.3 Hz, C–F, CH), 127.7 (CH), 124.7 (d, *J* = 2.8 Hz, C–F, CH), 122.9 (d, *J* = 14.6 Hz, C–F), 120.3, 118.4 (d, *J* = 23.6 Hz, C–F, CH), 115.6 (d, *J* = 21.0 Hz, C–F, CH), 111.8 (CH), 108.1 (d, *J* = 26.7 Hz, C–F, CH), 61.0 (CH_2_), 37.4 (d, *J* = 3.4 Hz, C–F, CH_2_). Found: C, 59.60; H, 3.82; Cl, 7.96; F, 8.51; N, 12.58. Anal. calcd (%) for C_22_H_17_ClF_2_N_4_O_2_: C, 59.67; H, 3.87; Cl, 8.00; F, 8.58; N, 12.65.

1-(2-(2-(5-Fluoro-1-(6-chloro-2-fluorobenzyl)-2-oxoindolin-3-ylidene)hydrazinyl)-2-oxoethyl)pyridin-1-ium chloride (**5b**). Yellow powder. Yield 83%, m.p. = 152–153 °C. IR spectrum, ν, cm^−1^: 1355 (C–F), 1634 (C=C), 1687 (C=O), 3017 (CH), 3393 (NH). ^1^H NMR (500 MHz, DMSO-*d*_6_) δ 12.63 (s, 1H, NH), 9.10 (d, *J* = 6.1 Hz, 2H, Ar), 8.73 (t, *J* = 7.6 Hz, 1H, Ar), 8.27 (t, *J* = 6.8 Hz, 2H, Ar), 7.48–7.34 (m, 4H, Ar), 7.31–7.26 (m, 1H, Ar), 7.09–7.07 (m, 1H, Ar), 6.22 (s, 2H, CH_2_). 5.15 (s, 2H, CH_2_). ^13^C NMR (126 MHz, DMSO-*d*_6_) δ 167.5, 163.5, 161.3 (d, *J* = 249.2 Hz, C–F), 158.6 (d, *J* = 239.8 Hz, C–F), 146.6 (CH), 146.5 (CH), 146.2, 139.4, 134.3 (d, *J* = 5.0 Hz, C–F, CH), 130.0 (d, *J* = 9.8 Hz, C–F, CH), 127.8 (CH), 127.7 (CH), 125.9 (d, *J* = 2.9 Hz, C–F), 120.2 (d, *J* = 16.5 Hz, C–F), 118.4 (d, *J* = 24.3 Hz, C–F, CH), 114.9 (d, *J* = 22.6 Hz, C–F, CH), 111.5 (d, *J* = 13.9 Hz, C–F), 108.1 (d, *J* = 2.3 Hz, C–F, CH), 61.1 (CH_2_), 36.0 (d, *J* = 1.7 Hz, C–F, CH_2_). Found: C, 55.30; H, 3.39; Cl, 14.79; F, 7.86; N, 11.70. Anal. calcd (%) for C_22_H_16_Cl_2_F_2_N_4_O_2_: C, 55.36; H, 3.38; Cl, 14.85; F, 7.96; N, 11.74.

1-(2-(2-(5-Fluoro-2-oxo-1-(3-(trifluoromethyl)benzyl)indolin-3-ylidene)hydrazinyl)-2-oxoethyl)pyridin-1-ium chloride (**5c**). Yellow powder. Yield 79%, m.p. = 172–173 °C. IR spectrum, ν, cm^−1^: 1330 (C–F), 1637 (C=C), 1702 (C=O), 1680 (C=O), 3018 (CH), 3392 (NH). ^1^H NMR (500 MHz, DMSO-*d*_6_) δ 9.04 (d, *J* = 5.9 Hz, 2H, Ar), 8.71 (t, *J* = 7.8 Hz, 1H, Ar), 8.23 (t, *J* = 6.6 Hz, 2H, Ar), 7.74 (s, 1H, Ar), 7.68–7.65 (m, 2H, Ar), 7.61 (d, *J* = 7.8 Hz, 1H, Ar), 7.56–7.54 (m, 1H, Ar), 7.31–7.27 (m, 1H, Ar), 7.12 (dd, *J* = 8.8 Hz, *J* = 3.9 Hz, 1H, Ar), 6.16 (s, 2H, CH_2_), 5.09 (s, 2H, CH_2_). ^13^C NMR (126 MHz, DMSO-*d*_6_) δ 168.0, 161.0, 159.2 (d, *J* = 239.8 Hz, C–F), 147.1 (CH), 146.7 (CH), 135.0, 131.8 (CH), 130.4 (CH), 129.9 (q, *J* = 31.7 Hz, C–F), 128.2 (CH), 125.0 (m, CH), 124.4 (m, CH), 124.2 (q, *J* = 272.4 Hz, C–F), 120.8, 118.8 (d, *J* = 23.7 Hz, C–F, CH), 112.2–121.1 (m, 2C, 2CH), 108.8 (d, *J* = 25.3 Hz, C–F, CH), 61.1 (CH_2_), 42.7 (CH_2_). ^19^F NMR (600 MHz, DMSO-*d*_6_) δ −119.3, −61.0. Found: C, 56.00; H, 3.47; Cl, 7.17; F, 15.36; N, 11.38. Anal. calcd (%) for C_23_H_17_ClF_4_N_4_O_2_: C, 56.05; H, 3.48; Cl, 7.19; F, 15.42; N, 11.37.

1-(2-(2-(5-Fluoro-2-oxo-1-((perfluorophenyl)methyl)indolin-3-ylidene)hydrazinyl)-2-oxoethyl)pyridin-1-ium chloride (**5d**). Yellow powder. Yield 72%, m.p. = 165–166 °C. IR spectrum, ν, cm^−1^: 1337 (C–F), 1636 (C=C), 1707 (C=O), 1731 (C=O), 3030 (CH), 3402 (NH). ^1^H NMR (600 MHz, DMSO-*d*_6_) δ 12.59 (s, 1H, NH), 9.17–9.13 (m, 2H, Ar), 8.74 (t, *J* = 7.0 Hz, 1H, Ar), 8.28–8.26 (m, 2H, Ar), 7.51–7.48 (m, 1H, Ar), 7.43–7.40 (m, 1H, Ar), 7.30–7.28 (m, 1H, Ar), 6.26 (s, 2H, CH_2_). 5.16 (s, 2H, CH_2_). ^13^C NMR (151 MHz, DMSO-*d*_6_) δ 167.7, 160.2, 158.7 (d, *J* = 239.8 Hz, C–F), 146.6 (CH), 146.5 (CH), 145.0 (dm, *J* = 248.2 Hz, C–F), 140.4 (dm, *J* = 252.0 Hz, C–F), 136.9 (dm, *J* = 249.4 Hz, C–F), 133.8, 127.6 (CH), 127.5 (d, *J* = 27.4 Hz, C–F, CH), 120.2, 118.5 (d, *J* = 24.0 Hz, C–F, CH), 111.6, 109.1, 107.9 (d, *J* = 26.1 Hz, C–F, CH), 61.0 (CH_2_), 32.0 (CH_2_). Found: C, 51.30; H, 2.50; Cl, 6.87; F, 22.08; N, 10.80. Anal. calcd (%) for C_22_H_13_ClF_6_N_4_O_2_: C, 51.33; H, 2.55; Cl, 6.89; F, 22.14; N, 10.88.

1-(2-(2-(5-Methyl-2-oxo-1-(3-(trifluoromethyl)benzyl)indolin-3-ylidene)hydrazinyl)-2-oxoethyl)pyridin-1-ium chloride (**5e**). Yellow powder. Yield 85%, m.p. = 202–203 °C. IR spectrum, ν, cm^−1^: 1375 (C–F), 1624 (C=C), 1688 (C=O), 3011 (CH), 3402 (NH). ^1^H NMR (500 MHz, DMSO-*d*_6_) δ 12.70 (s, 1H, NH), 9.14 (d, *J* = 5.7 Hz, 2H, Ar), 8.74 (t, *J* = 7.7 Hz, 1H, Ar), 8.28 (t, *J* = 7.2 Hz, 2H, Ar), 7.79 (s, 1H, Ar), 7.67–7.66 (m, 2H, Ar), 7.47 (s, 1H, Ar), 7.61 (d, *J* = 7.4 Hz, 1H, Ar), 7.47 (s, 1H, Ar), 7.28 (d, *J* = 7.3 Hz, 1H, Ar), 7.07 (d, *J* = 8.0 Hz, 1H, Ar), 6.25 (s, 2H, CH_2_). 5.11 (s, 2H, CH_2_), 2.33 (s, 3H, CH_3_). ^13^C NMR (126 MHz, DMSO-*d*_6_) δ 164.2, 163.9, 153.2, 146.6 (CH), 146.5 (CH), 140.8, 137.2, 135.2, 132.8 (CH), 132.5, 131.4, 129.8 (CH), 129.4 (q, *J* = 31.9 Hz, C–F), 127.7 (CH), 127.6 (CH), 124.4, 124.1 (CH), 124.0 (q, *J* = 273.9 Hz, C–F), 121.1, 110.4 (CH), 61.0 (CH_2_), 42.2 (CH_2_), 20.4 (CH_3_). ^19^F NMR (600 MHz, DMSO-*d*_6_) δ −61.0. Found: C, 58.90; H, 4.09; Cl, 7.27; F, 11.57; N, 11.43. Anal. calcd (%) for C_24_H_20_ClF_3_N_4_O_2_: C, 58.96; H, 4.12; Cl, 7.25; F, 11.66; N, 11.46.

### 3.2. Biological Studies

#### 3.2.1. Anticancer Activity

##### Cell Lines and Their Cultivation

Human cell cultures of tumor origin—HeLa (cervical tumor), HuTu 80 (human duodenal adenocarcinoma), ChangLiver (human liver HeLa-like line), and normal cell line WI38, VA 13 subline 2RA is a diploid human cell strain composed of fibroblasts derived from healthy lung tissue provided by The Gamaleya National Center for Epidemiology and Microbiology, as well as the Institute of Cytology of the Russian Academy of Sciences. They were grown in a nutrient medium DMEM (PanEco, Moscow, Russia) and MEM (PanEco, Moscow, Russia), containing fetal bovine serum (10% by volume) (ThermoFisher Scientific, Paisley, UK), Glutamax (2 mM) (Gibco, Scotland, UK), and penicillin-streptomycin (1% by volume) (PanEco, Moscow, Russia). The cultivation was carried out at 37 °C in a humidified CO_2_ atmosphere (5%).

##### Determination of Cell Viability

Cell viability was determined by the MTT test [63]. Cells were seeded in a 96-well plate in the amount of 1 × 10^4^ cells/200 µL of complete nutrient medium and cultured at 37 °C in CO_2_ (5%). After 24 h of incubation, various concentrations of test compounds in the range of 0.1 to 100 μM were added to the cell cultures, and then the cells were cultured under the same conditions for 24 and 72 h. For each concentration, the experiment was carried out in triplicate. 

All compounds were dissolved in DMSO and then diluted with the medium to the required concentration. The final content of DMSO in the well did not exceed 1% and did not have a toxic effect on the cells. The DMSO was also added to the control wells in a volume of 1%. After the incubation time, MTT (3-(4,5-dimethylthiazol-2-yl)-2,5-diphenyltetrazolium bromide, 5 mg/mL) was added to each well, and plates were additionally incubated for 2 h (until the characteristic color appeared).

Using a plate analyzer (Cytation3, BioTech Instruments Inc., Winooski, VT, USA), the optical density was determined at 530 nm. The concentration value causing 50% inhibition of cell population growth (IC_50_) was determined from dose-dependent curves.

The IC_50_ parameter was calculated using the AAT Bioquest program [64].

The selectivity index (SI) is calculated as the ratio between the IC_50_ for normal cells and the IC_50_ for cancer cells [65].

##### Apoptosis

HuTu 80 cells at 1 × 10^6^ cells/well in a final volume of 2 mL were seeded into six-well plates. After 24 h of incubation, various concentrations of tested compounds were added to the wells. The cells were harvested at 2757 g for 5 min and then washed twice with ice-cold PBS, followed by resuspension in binding buffer. Next, the samples were incubated with 5 μL of annexin V-Alexa Fluor 647 (Sigma-Aldrich, Burlington, MA, USA) and 5 μL of propidium iodide for 15 min at room temperature in the dark. Finally, the cells were analyzed by flow cytometry (Guava easyCyte, Merck, Rahway, NJ, USA) within 1 h. The experiments were repeated three times.

##### Mitochondrial Membrane Potential

To analyze the effect of compounds on changes in the mitochondrial membrane potential, we performed flow cytometry using JC-10 fluorescent dye (in the Mitochondria Membrane Potential Kit). In normal cells (with high membrane potential), JC-10 accumulates in the mitochondrial matrix, where it forms aggregates with red fluorescence. However, in apoptotic cells, JC-10 diffuses out of mitochondria, converts to its monomeric form, and emits green fluorescence, which is recorded by a flow cytometer [66]. Cells were harvested at 2757 g for 5 min and then washed twice with ice-cold PBS, followed by resuspension in JC-10 (10 µg/mL) and incubation at 37 °C for 10 min. Then, the cells were rinsed three times and suspended in PBS, and the JC-10 fluorescence was observed by flow cytometry (Guava easyCyte, Merck, Rahway, NJ, USA).

##### ROS

HuTu 80 cells were incubated with tested compounds at concentrations of IC_50_ for 24 h. ROS generation was investigated using a flow cytometry assay and the CellROX^®^ Deep Red flow cytometry kit. For this, HuTu 80 cells were harvested at 2000 rpm for 5 min and then washed twice with ice-cold PBS, followed by resuspension in 0.1 mL of medium without FBS, to which was added 0.2 μL of CellROX^®^ Deep Red and incubated at 37 °C for 30 min. After three times washing the cells and suspending them in PBS, the production of ROS in the cells was immediately monitored using a flow cytometer (Guava easyCyte, Merck, Rahway, NJ, USA).

##### Anticoagulant and Antiaggregation Activities Study

The in vitro experiments were performed using the blood of healthy male donors aged 18–24 years (a total of 54 donors). This study was approved by the Ethics Committee of the Federal State Budgetary Educational Institution of Higher Education at the Bashkir State Medical University of the Ministry of Health of the Russian Federation (No. 2, dated 17 October 2012). Informed consent was obtained from all participants before blood sampling. The blood was collected from the cubital vein using the vacuum blood collection system BD Vacutainer^®^ (Becton, Dickinson, and Company, Franklin Lakes, NJ, USA). A 3.8% sodium citrate solution in a 9:1 ratio was used as a venous blood stabilizer. The study of the effect on platelet aggregation was performed using the Born method [67] using the aggregometer «AT-02» (SPC Medtech, Moscow, Russia). The assessment of the antiplatelet activity of the studied compounds and reference preparations was started with a final concentration of 2 × 10^−3^ mol/L. Adenosine diphosphate (ADP; 20 μg/mL) and collagen (5 mg/mL) manufactured by Tehnologia-Standart Company, Barnaul, Russia, were used as inducers of aggregation. The study on the anticoagulant activity was performed by standard recognized clotting tests using the optical two-channel automatic analyzer of blood coagulation, Solar CGL 2110 (CJSC SOLAR, Minsk, Belarus). The following parameters were studied: activated partial thromboplastin time (APTT), prothrombin time (PT), and fibrinogen concentrations according to the Clauss method. The determination of the anticoagulant activity of the studied compounds and reference preparation was performed at a concentration of 5 × 10^−4^ g/mL using the reagents manufactured by Tehnologia-Standart Company (Barnaul, Russia). The results of the study were processed using the statistical package Statistica 10.0 (StatSoft Inc., Tulsa, OK, USA). The Shapiro–Wilk’s test was used to check the normality of the actual dataset distribution. The form of distribution of the data obtained differed from the normal one; therefore, non-parametric methods were used for further analysis. The data were presented as medians and 25 and 75 percentiles. An analysis of variance was conducted using the Kruskal–Wallis test. A *p*-value of 0.05 was considered statistically significant.

#### 3.2.2. Antiphytopathogenic Activity

Plant pathogenic bacterial strains *Micrococcus luteus* B-109, *Pectobacterium atrosepticum* 1043, *Pectobacterium carotovorum* subsp. *carotovorum* MI, *Pseudomonas fluorescens* EL-2.1, and *Xanthomonas campestris* B-610 and the fungal strain *Fusarium oxysporum* IBPPM 543 were from the Collection of Rhizosphere Microorganisms, IBPPM RAS (WFCC no. 975, WDCM no. 1021) (CM IBPPM, http://collection.ibppm.ru). The pathogenic fungus *Phytophthora cactorum* VKM F-985, provided by the All-Russian Collection of Microorganisms (VKM) and deposited at the A.E. Favorsky Irkutsk Institute of Chemistry, SB RAS, was also used as a test organism. Bacteria *M. luteus*, *P. carotovorum* subsp. *carotovorum*, *P. atrosepticum*, and *P. fluorescens* were grown in a meat–peptone medium (BP), and X. campestris was grown in the medium with glucose, yeast extract, and calcium carbonate (GYCa). Solid media contained Bacto agar (18 g/L); pH was adjusted to 7.2–7.4. All bacterial cultures were grown at 28 °C. The mycelial cultures of *F. oxysporum* and *P. cactorum* were grown on a glucose–peptone–yeast (GPY) nutrient medium at 27 °C. For inoculum preparation, both fungal strains were initially grown on the agar GPY in Petri dishes and then transferred into the seed medium by punching out 5 mm of the agar plate culture with a self-designed cutter.

The antibacterial and antifungal activities of the compounds were explored using the agar-well diffusion method and the technique of mycelial radial growth inhibition. The method of diffusion in agar (measuring the diameter of growth inhibition zones) was used to determine the bactericidal activity. The 6-mm wells were made in agar medium (GYCa for *Xanthomonas campestris* or BP for other bacteria). Bacterial suspensions were distributed over the agar surface, and the tested compound’s solution (150 μL) was added to each well. The width of growth inhibition zones around the wells was determined after incubation for 36–40 h.

For the fungicidal activity analysis, radial growth (colony diameters) of the fungi on a solid medium in the absence and in the presence of the compounds **5a**–**e** and **7a**–**c** solutions at various concentrations were compared. The method consisted of the following: A sterile GPY agar medium (20.0 mL), which was melted and then allowed to cool to approximately 60 °C, was mixed with the precisely measured volumes of the solutions under question and poured into a sterile Petri dish (90 mm i.d.). After the solidification of GPY agar, the media were inoculated by the fungus using 10-day-old cultures of *F. oxysporum* or *P. cactorum*. The inoculation was conducted by transferring a 5-mm (i.d.) GPY-agar block covered with mycelium to the center of the Petri dish, followed by incubation in a thermostat at 27 °C. The fungicidal effect was scored by the size of the mycelium colony on a Petri dish compared to the control without fungicidal admixtures to GPY agar. Each treatment was performed in at least four replicates in two independent experiments. The observation period ended when the control Petri dish was filled with mycelium (usually after 12 days). The inhibition of the phytopathogen colony growth by the compounds or the reference fungicide solutions was calculated as a percentage by which the mycelium radial propagation was decreased compared to the unaffected control; the latter was taken as 100% of growth (or zero percent of inhibition). The EC_50_ value was calculated as the compound concentration at which the radial growth of the fungus colony was decreased by 50% relative to the non-fungicidal control. This effective concentration causing 50% of the fungal growth inhibition was determined from the dose-relative inhibition curves plotted for each tested compound. The solutions of the compounds were prepared with a concentration of 2 mmol/L (stock solution). For comparison, test compounds were also commonly applied as disinfecting agents, and antibiotics were used as a positive control.

#### 3.2.3. Statistical Analysis

The data were expressed as the mean ± SEM. Statistical comparisons were made using one-way analysis of variance (ANOVA), followed by Dunnett’s multiple comparison tests. Two-way repeated measures (mixed model) ANOVA followed by Bonferroni posttests were also used to compare the recognition of two objects. A difference with a *p*-value ≤ 0.05 was considered statistically significant. The statistical analysis was performed using GraphPad Prism 5 (GraphPad Software, San Diego, CA, USA).

## 4. Conclusions

A series of novel fluorinated 1-benzylisatins was synthesized with high yields by an easy one-pot reaction procedure. Using the acid-catalyzed reaction of new isatins with Girard’s reagent P and its dimethyl analog, a wide range of water-soluble isatin-3-hydrazones were obtained. The most promising use of modified isatins as a basis for the creation of anticancer agents is possessed by ortho-substituted fluorine (**3a**), chlorine (**3b**), or both (**3d**) atoms in the benzyl fragment of the compound. The mechanism of the cytotoxic action of these substances is associated with apoptosis induction due to mitochondrial membrane dissipation and stimulation of reactive oxygen species production in tumor cells. It was shown that compounds **3a** and **3b** exhibit antiaggregation activity at the level of acetylsalicylic acid, and the whole series of fluorine-containing isatins does not adversely affect the hemostasis system as a whole. In a wide range of new water-soluble pyridinium isatin-3-acylhydrazones, substances **7c** and **5c**,**e** exhibit the highest activity. They have an antagonistic effect against phytopathogens of bacterial and fungal origin and can be considered biological preparations for combating plant diseases.

## Data Availability

Data is contained within the article.

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
