# Peer review of "Anticancer and Antiphytopathogenic Activity of Fluorinated Isatins and Their Water-Soluble Hydrazone Derivatives"

_ijms, 2023, doi:10.3390/ijms242015119_

Round 1

Reviewer 1 Report

The article by Bogdanov et al. describes the synthesis of derivatives of fluorinated isatins and the evaluation of their anticancer/antibacterial/antifungal properties. Most importantly, the evidence supports most of the claims made in the "Results and Discussion" and "Conclusions" sections. Together with the novelty of the article, it can be published. However, there are issues that require some attention.

General remarks

1.       Some parts of the "Results and discussion" section belong to the "Materials and methods" section, because they describe the methodology rather than the results and their interpretation. For example - SI, IC50, EC50 calculation.

2.       I know that IJMS allows to merge "Results" and "Discussion" sections, but I strongly encourage to separate results from their interpretation. In my opinion, the merged sections greatly reduced the clarity of this article. Moreover, large parts of this section should be moved to "Introduction", as they are background information rather than interpretation of the results.

Cytotoxicity study

1.       Why did the authors use cervical cancer, duodenal cancer and normal liver cell lines? To me it's quite irrational, especially since the authors used tamoxifen - which is an anti-estrogenic rather than an "antitumor" drug (which should be underlined in the manuscript). To the best of my knowledge, TAM is used in the therapy of breast cancer, and reports on TAM and cervix rather concern side effects of TAM therapy (e.g. https://www.ncbi.nlm.nih.gov/pmc/articles/PMC2482152/). The use of TAM against duodenal cancer seems a bit ill-conceived. Unless the authors somehow explain their intentions.
Evaluation of liver cell line toxicity is quite reasonable (due to TAM liver toxicity), but Chang Liver cell line may not be the best choice - as it may be contaminated with HeLa cells (as described here: https://journals.lww.com/hep/Fulltext/2011/11000/Cell_line_misidentification__The_case_of_the_Chang.51.aspx and here https://www.atcc.org/products/ccl-13). It simply makes some of the claims made in the article at least doubtful, as this cell line cannot be considered "normal". Unless the authors provide evidence that their Chang Liver cell line is free of contamination from other cell types.

Furthermore, I would advise against the calculation of SI - since all cell lines (besides the normal/cancer difference) represent completely different origins and cell types, the calculation of SI may be biased.

The use of 5-FU is not a concern as it is a much more versatile drug than TAM.

2.       In Material and Methods, the authors describe the concentration used as "from 0.1 to 100 µM", but do not state how many different concentrations they actually tested. This is especially important since they estimated the IC50 and it's precision is highly dependent on the number of concentrations tested. Also, cytotoxicity curves should be shown, at least in supplementary files.

Mitochondrial membrane potential evaluation

1.             As far as I can remember, both major apoptotic pathways (intrinsic and extrinsic) are closely involved in mitochondria. Even though the extrinsic pathway relies on external stimuli, it eventually leads to permeabilization of the mitochondrial outer membrane and loss of membrane potential [Kiraz, Yağmur, et al. "Major apoptotic mechanisms and genes involved in apoptosis." Tumor Biology 37 (2016): 8471-8486.]. Therefore, analysis of JC-10 does not provide insight into which pathway (intrinsic or extrinsic) is activated. It's just the evidence that apoptosis is occurring.

2.             "Normal functioning" of mitochondria does not depend on "permeability of mitochondrial membrane". It depends on the impermeability of the inner membrane. Highly impermeable, cardiolipin-rich membrane of mitochondria allows to maintain the potential necessary for functioning of ATP synthase.

Figures

1.       Figure 1: The name of the first connection is missing

2.       Figure 2: It's not clear whether the middle compound belongs to the antianthropopathogenic group or to the antianthropo/antiphytopathogenis group.

3.       Figure 5: A and B labels should be on the upper left side of each compound - or at least below or on the left side, not on the right side.

4.       Figures 8, 9 and 10: What is the p-value behind the  asterisks? It should be given in the figure caption, along with the test used.

Materials and methods

1.       The subsection "Statistical analysis" is missing. Also, all tests used to analyze different results should be mentioned in the figure captions.

2.       Is it really important to include so much detail in the "Chemistry" subsection? I admire the meticulousness of the authors, but I would ask the editors to allow a large part of this section to be included in the Supplementary files.

Author Response

1 Reviewer

The article by Bogdanov et al. describes the synthesis of derivatives of fluorinated isatins and the evaluation of their anticancer/antibacterial/antifungal properties. Most importantly, the evidence supports most of the claims made in the "Results and Discussion" and "Conclusions" sections. Together with the novelty of the article, it can be published. However, there are issues that require some attention.

Response: Thank you for this positive evaluation of our work!

General remarks

  1. Some parts of the "Results and discussion" section belong to the "Materials and methods" section, because they describe the methodology rather than the results and their interpretation. For example - SI, IC50, EC50 calculation.

Answer.           Thank you very much for your comment. Some details of calculating various parameters have been moved to the "Materials and methods" section.

  1. I know that IJMS allows to merge "Results" and "Discussion" sections, but I strongly encourage to separate results from their interpretation. In my opinion, the merged sections greatly reduced the clarity of this article. Moreover, large parts of this section should be moved to "Introduction", as they are background information rather than interpretation of the results.

Answer.           The authors thank the reviewer for this remark and fully agree with it. However, within the framework of this work, unfortunately, we will not be able to divide the results and discussion section into two sections. Most importantly, there are several directions in the study of biological activity that cannot be discussed in a single context. At the moment, the manuscript follows the logic of separately studying the antitumor activity for isatin derivatives and the antiphytopathogenic properties for hydrazones. The  division of the results and discussion section will disrupt the built-in logic of the work and lead to less efficient presentation of our findings.

Cytotoxicity study

  1. Why did the authors use cervical cancer, duodenal cancer and normal liver cell lines? To me it's quite irrational, especially since the authors used tamoxifen - which is an anti-estrogenic rather than an "antitumor" drug (which should be underlined in the manuscript). To the best of my knowledge, TAM is used in the therapy of breast cancer, and reports on TAM and cervix rather concern side effects of TAM therapy (e.g. https://www.ncbi.nlm.nih.gov/pmc/articles/PMC2482152/). The use of TAM against duodenal cancer seems a bit ill-conceived. Unless the authors somehow explain their intentions.
    Evaluation of liver cell line toxicity is quite reasonable (due to TAM liver toxicity), but Chang Liver cell line may not be the best choice - as it may be contaminated with HeLa cells (as described here: https://journals.lww.com/hep/Fulltext/2011/11000/Cell_line_misidentification__The_case_of_the_Chang.51.aspx and here https://www.atcc.org/products/ccl-13). It simply makes some of the claims made in the article at least doubtful, as this cell line cannot be considered "normal". Unless the authors provide evidence that their Chang Liver cell line is free of contamination from other cell types.

Furthermore, I would advise against the calculation of SI - since all cell lines (besides the normal/cancer difference) represent completely different origins and cell types, the calculation of SI may be biased.

The use of 5-FU is not a concern as it is a much more versatile drug than TAM.

Answer.           Indeed, the cell lines used in this work have different origins. Initially, the task was to look at the effect of the studied compounds on different tumor cells and compare their effect with the effect on the survival of a conditionally normal cell line - Chang liver. In the future development of the work, we will follow with great pleasure the recommendation of a respected reviewer and add the proposed cell lines for comparison.

The Chang liver line was originally derived from healthy human liver tissue. It was shown that cells of this line synthesize proteins characteristic of human liver cells, in particular, hepatic-type alkaline phosphatase, unlike HeLa cells, and can be used as a conditionally normal cell line to compare the effects of substances on the tumor and healthy microenvironment. (Ludueña MA, Iverson GM, Sussman HH. Expression of liver and placental alkaline phosphatases in Chang liver cells. J Cell Physiol. 1977 Apr;91(1):119-29. doi: 10.1002/jcp.1040910112.)

We also agree with the reviewer that 5-FU is more appropriate as a reference drug in Table 1. TAM was removed from the results of the manuscript.

The selectivity index allows the authors in this work to select some of the most promising compounds for further chemical modification in order to obtain the most cytotoxic substances in relation to tumor cells and the least harm to a healthy microenvironment

  1. In Material and Methods, the authors describe the concentration used as "from 0.1 to 100 µM", but do not state how many different concentrations they actually tested. This is especially important since they estimated the IC50 and it's precision is highly dependent on the number of concentrations tested. Also, cytotoxicity curves should be shown, at least in supplementary files.

Answer.           Ten concentrations were tested for each compound. Cytotoxicity curves are given in supplementary materials.

Mitochondrial membrane potential evaluation

  1. As far as I can remember, both major apoptotic pathways (intrinsic and extrinsic) are closely involved in mitochondria. Even though the extrinsic pathway relies on external stimuli, it eventually leads to permeabilization of the mitochondrial outer membrane and loss of membrane potential [Kiraz, Yağmur, et al. "Major apoptotic mechanisms and genes involved in apoptosis." Tumor Biology 37 (2016): 8471-8486.]. Therefore, analysis of JC-10 does not provide insight into which pathway (intrinsic or extrinsic) is activated. It's just the evidence that apoptosis is occurring.

Answer.           We fully agree with the comment of the distinguished reviewer and revised the discussion to reflect it.

  1. "Normal functioning" of mitochondria does not depend on "permeability of mitochondrial membrane". It depends on the impermeability of the inner membrane. Highly impermeable, cardiolipin-rich membrane of mitochondria allows to maintain the potential necessary for functioning of ATP synthase.

Answer.           Thank you to the reviewer for such a helpful remark. We corrected this phrase in the text of the manuscript.

Figures

  1. Figure 1: The name of the first connection is missing

Answer.           Thank you for your comment. In this form, we tried to give an example of isatin derivatives that exhibit antibacterial properties. The first compound in Figure 1 does not have a common name, so we have applied a term found in the literature to its name (Zhi Xu, Shi-Jia Zhao, Zao-Sheng Lv, Feng Gao et al., Eur J Med Chem, 2019, 162, 396-406. doi: 10.1016/j.ejmech.2018.11.032).

  1. Figure 2: It's not clear whether the middle compound belongs to the antianthropopathogenic group or to the antianthropo/antiphytopathogenis group.

Answer.           Thank you for your valuable comment. This figure was modified to remove the ambiguity.

  1. Figure 5: A and B labels should be on the upper left side of each compound - or at least below or on the left side, not on the right side.

Answer.           After changes suggested by another reviewer, the number of this figure became 4. Labels were added to the upper left corner of the figures.

  1. Figures 8, 9 and 10: What is the p-value behind the asterisks? It should be given in the figure caption, along with the test used.

Answer.           We thank the reviewer for the helpful remark and apologize for omitting this information to the figure captions. We have corrected this error.

Materials and methods

  1. The subsection "Statistical analysis" is missing. Also, all tests used to analyze different results should be mentioned in the figure captions.

Answer.           We thank the reviewer for the comment. The section "Statistical analysis" has been added to "materials and methods".

  1. Is it really important to include so much detail in the "Chemistry" subsection? I admire the meticulousness of the authors, but I would ask the editors to allow a large part of this section to be included in the Supplementary files.

Answer.           This presentation of the "Chemistry" section was made on the basis of the "Guide for Authors" and in accordance with the reference to the most recent articles of the IJMS journal (for example: Ivan, B.-C.; Barbuceanu,S.-F.; Hotnog, C.M.; et al., Synthesis, Characterization and Cytotoxic Evaluation of New Pyrrolo[1,2-b]pyridazines Obtained via Mesoionic Oxazolo-Pyridazinones. Int. J. Mol. Sci. 2023, 24, 11642. https://doi.org/10.3390/ijms241411642; Hasnowo, L.A.; Larkina, M.S.; Plotnikov, E.; Bodenko, V.; et al. Synthesis, 123I-Radiolabeling Optimization, and Initial Preclinical Evaluation of Novel Urea-Based PSMA Inhibitors with a Tributylstannyl Prosthetic Group in Their Structures. Int. J. Mol. Sci. 2023, 24, 12206. https://doi.org/10.3390/ijms241512206). It seems to us important to give readers sufficient details about the equipment used in the work, the general procedures for the synthesis of new compounds and their physicochemical characteristics.

Thank you very much for this detailed and helpful analysis!

Reviewer 2 Report

I think this paper would be of interest to a graduate student wanting to see some straightforward organic chemistry with some boiler plate assays. The new compounds are not designed with a target in mind, their potency and selectivity are low and will e metabolically (CYPs) labile. This may be suitable for a lower impact journal, but it'd be up to the editors and reviewers there. This reviewer believes this study is not high impact enough for publication in IJMS. 

The English is in good shape! Good Job. 

Author Response

2 Reviewer

I think this paper would be of interest to a graduate student wanting to see some straightforward organic chemistry with some boiler plate assays. The new compounds are not designed with a target in mind, their potency and selectivity are low and will be metabolically (CYPs) labile. This may be suitable for a lower impact journal, but it'd be up to the editors and reviewers there. This reviewer believes this study is not high impact enough for publication in IJMS.

Answer.           We appreciate the frankness of this reviewer. Unfortunately, the reviewer offered no specific concerns besides his/her general opinion that organic chemistry is straightforward and that the biological assays are well-known. However, the value of this work is in the combination of the two and the discovery of the initial leads that can guide the future path to more potent compounds. In general, one cannot take for granted any new findings of specific biological activity from a new set of molecules.

The reviewer assumes that the compounds will be metabolically labile. We certainly agree that one should always be concerned about the metabolic profile. This is true for any new set of molecules but, in this particular case, there are ample precedents of structurally related isatins to have suitable metabolic profiles. Furthermore, fluorination is known to greatly increase metabolic stability by providing greater resistance to oxidation.

We are glad that the two other reviewers, who did provide detailed feedback, did not share this opinion.

Reviewer 3 Report

Reviewers' comments (Manuscript ID# ijms-2569665)

The manuscript by Bogdanov describes the anticancer and antiphytopathogenic activity of fluorinated isatins and their water-soluble hydrazone derivatives. Author has synthesized series of benzylated isatin derivatives in good yield and screened for their cytotoxicity against two cancer cell lines (M-HeLa & HuTu-80) and one normal human hepatocyte. Some of the synthesized compounds showed moderate activity with moderate selectivity. Furthermore, the cell death mechanism associated with apoptosis due to mitochondrial membrane dissipation and stimulation of ROS species production in tumor cells. In addition to this, author has prepared few water soluble pyridinium isatin hydrazine derivates and which shows antagonistic effect against phytopathogens of bacterial and fungal origin. This reviewer is recommending this manuscript for the publication in IJMS after minor revision.

Comments:

1.      Scheme 1 is one pot or two step reaction?

2.      In fig.3 please provide appropriate number or name.

3.      For Table 1 caption, author should provide the incubation time.

4.      Is there any reason why author did not evaluate the compounds 7a-c for their anticancer activity?

5.      General synthetic procedure for the synthesis of 3a-f and 4a-e can be rewritten, it is quite unclear to reader.

6.      Author has prepared several fluorinated isatin derivatives and all are well characterized. However, 19F NMR data is missing in the experimental. Author can provide 19F NMR data for at least active compounds.

Author can check this manuscript for typo error before revision. 

Quality of English Language is good except minor spell check and typo errors...

Author Response

3 Reviewer

The manuscript by Bogdanov describes the anticancer and antiphytopathogenic activity of fluorinated isatins and their water-soluble hydrazone derivatives. Author has synthesized series of benzylated isatin derivatives in good yield and screened for their cytotoxicity against two cancer cell lines (M-HeLa & HuTu-80) and one normal human hepatocyte. Some of the synthesized compounds showed moderate activity with moderate selectivity. Furthermore, the cell death mechanism associated with apoptosis due to mitochondrial membrane dissipation and stimulation of ROS species production in tumor cells. In addition to this, author has prepared few water soluble pyridinium isatin hydrazine derivates and which shows antagonistic effect against phytopathogens of bacterial and fungal origin. This reviewer is recommending this manuscript for the publication in IJMS after minor revision.

Response: Thank you very much for the helpful summary and positive evaluation of this work!

  1. Scheme 1 is one pot or two step reaction?

Answer.           The reaction shown in Scheme 1 is a two-step one-pot as the intermediate sodium salt undergoes further alkylation without isolation.

  1. In fig.3 please provide appropriate number or name.

Answer.           As recommended by the reviewer, Figures 3 and 4 have been revised.

  1. For Table 1 caption, author should provide the incubation time.

Answer.           As recommended by the reviewer, Table 1 caption has been corrected.

  1. Is there any reason why author did not evaluate the compounds 7a-c for their anticancer activity?

Answer.           We have previously studied the anticancer activity of these compounds. However, these hydrazones and some of their analogs did not show such activity; therefore, it was not studied in this work. At the same time, we have shown that isatin ammonium hydrazones have a high potential as compounds exhibiting antimicrobial activity against anthropo- and phytopathogens. Therefore, the anticancer activity of hydrazones 7а-с was not studied in this article.

  1. General synthetic procedure for the synthesis of 3a-f and 4a-e can be rewritten, it is quite unclear to reader.

Answer.           As recommended by the reviewer, this paragraph has been more detailed.

  1. Author has prepared several fluorinated isatin derivatives and all are well characterized. However, 19F NMR data is missing in the experimental. Author can provide 19F NMR data for at least active compounds.

Answer.           As recommended by the reviewer, this data have been more detailed.

Author can check this manuscript for typo error before revision.

Answer.           The manuscript has been double checked for typos and errors.

Round 2

Reviewer 1 Report

I am glad to see that the authors did their best to make the corrections for the article. However, I see that I should be more specific in my previous review, especially in my general remarks. For this round of review, I've attached a .pdf file with comments directly indicating all sentences that should be moved to the Introduction or Materials and methods sections, as well as several different remarks. Now I think it is unambiguous and as clear as possible.

Most importantly - I understand the authors' explanation as to why they prefer a merged "Results and discussion" section, and I fully accept their decision. But still, it is the "Results and Discussion" section, not the "Introduction, Materials and Methods, Results and Discussion" section. Background information and methodology need to be separated from the results and their interpretation for the sake of manuscript clarity, which (for now) is lacking in this article.

If the authors feel that after all these additions the Introduction would be too large and complex - adding several subsections to the Introduction would allow to maintain its clarity.

Regarding the Chang cell line - the article on which the authors base their claims was published in 1977. 46 years ago. At that time, no one was aware of the flaws of this cell line. Methods used in cited article could not find and contaminations. I agree that the cells in this cell line are mostly liver cells, but that's the point - "mostly". The contamination of the Chang cell line with HeLa has been proven and is a major drawback because even a small amount of cancer cells can significantly affect the biochemistry of normal cells. Because of this contamination, it's far from being a "healthy microenvironment".

Nevertheless, I agree that it does not disqualify this result, but this limitation must be included in this manuscript, the readers must be informed about it. I've indicated an appropriate place in the PDF, but if the authors want to put it elsewhere, that's fine too.

Also, some of the changes stated in the authors' response were not actually made in the manuscript. Especially in the part about the evaluation of the mitochondrial membrane potential. Please make sure not to forget them during this review round.

Author Response

Dear Mr. Kyrie Zhang,

We provide our detailed answers to the reviewer below.

Reviewer

1) I am glad to see that the authors did their best to make the corrections for the article. However, I see that I should be more specific in my previous review, especially in my general remarks. For this round of review, I've attached a .pdf file with comments directly indicating all sentences that should be moved to the Introduction or Materials and methods sections, as well as several different remarks. Now I think it is unambiguous and as clear as possible.

Most importantly - I understand the authors' explanation as to why they prefer a merged "Results and discussion" section, and I fully accept their decision. But still, it is the "Results and Discussion" section, not the "Introduction, Materials and Methods, Results and Discussion" section. Background information and methodology need to be separated from the results and their interpretation for the sake of manuscript clarity, which (for now) is lacking in this article.

If the authors feel that after all these additions the Introduction would be too large and complex - adding several subsections to the Introduction would allow to maintain its clarity.

Answer.           Thank you very much for your valuable comments. All comments kindly made for our convenience in the PDF file have been taken into account. In particular, the “Introduction” and “Results and Discussions” sections have been substantially revised in response to the reviewer's suggestions. This significantly reduced the volume of the text without changing the logic of the material and the essence of the manuscript.

Regarding the Chang cell line - the article on which the authors base their claims was published in 1977. 46 years ago. At that time, no one was aware of the flaws of this cell line. Methods used in cited article could not find and contaminations. I agree that the cells in this cell line are mostly liver cells, but that's the point - "mostly". The contamination of the Chang cell line with HeLa has been proven and is a major drawback because even a small amount of cancer cells can significantly affect the biochemistry of normal cells. Because of this contamination, it's far from being a "healthy microenvironment".

Nevertheless, I agree that it does not disqualify this result, but this limitation must be included in this manuscript, the readers must be informed about it. I've indicated an appropriate place in the PDF, but if the authors want to put it elsewhere, that's fine too.

Answer.           Chang liver cells have been shown to be functionally similar to normal liver cells [T. Yang, C. Li, L. Zhang, M. Li, and P. Zhou. A Promising Hepatocyte-Like Cell Line, CCL-13, Exhibits Good Liver Function Both In Vitro and in an Acute Liver Failure Model // Transplantation Proceedings, 45, 688 – 694 (2013) https://doi.org/10.1016/j.transproceed.2012.11.012]. In vitro Western blot analysis of Chang liver cells revealed the expression of ALB, UGT and CYP3A4. RT-PCR showed expression of functional markers enriched in liver, including albumin and GST mRNA. Regarding liver function markers and genes, no differences were observed between Chang liver cells and primary hepatocytes, indicating that the former have good metabolic and detoxifying functions. Laser confocal microscopy also revealed the expression of ALB, CYP3A4 and UGT in Chang liver cells. To study the general functions of the liver, Chang liver cells were transplanted inside the spleen, providing a protective effect. When injected into the spleen, Chang liver rats survived, exhibiting liver function shortly thereafter. We investigated the carcinogenic potential of Chang liver cells in nude mice. When 5 × 106 Chang liver cells were injected subcutaneously, nude mice were tumor-free within 1 month. In contrast, nude mice transplanted with the same number of HepG2 cells developed tumors within 2 weeks, indicating the low tumorigenicity of Chang liver cells.

3) Also, some of the changes stated in the authors' response were not actually made in the manuscript. Especially in the part about the evaluation of the mitochondrial membrane potential. Please make sure not to forget them during this review round.

Answer.           We thank the reviewer for this valuable comment. We have revised the conclusion in the part with mitochondrial potential and moved text to the methods section. We really hope that now our manuscript has become more understandable and easier for readers to follow.

Round 3

Reviewer 1 Report

I had hoped that this would be the last round of revisions, but it seems that minor revisions are still needed.

Despite what the authors claim, the Chang cell line cannot be used as a "normal" cell line. I mentioned why in the first and second review rounds and I am surprised that the authors still cannot accept this. It's not an opinion - it's a fact: The Chang line is contaminated with HeLa cells, which was established in 2001 [Masters JR et al. (2001) Proc Natl Acad Sci U S A 98(14):8012-7. http://www.pnas.org/content/98/14/8012.full] and as such cannot be used as a "normal" cell line. Even though it is sometimes used due to lack of knowledge of authors/reviewers/editors (as shown here: [https://iclac.org/wp-content/uploads/Chang-Liver_Case-Study_v1_3.pdf]), this cell line should definitely not be used.

Again, see https://journals.lww.com/hep/Fulltext/2011/11000/Cell_line_misidentification__The_case_of_the_Chang.51.aspx

And

- which comprehensively describes the history of this cell line, supported by all necessary evidence required.

In the study by Yang et al. (2013) that you cite, only the liver marker proteins were tested; which was obviously positive for the Chang liver cell line. But they did not check HeLa marker proteins or genes. Also, transplantation of this cell line into the liver, which you mentioned does not make this cell line credible. Liver-derived cells would certainly be better tolerated by the liver than cervical cancer cells. However, this does not make this cell line line valid as a "normal cell line" - in vitro they are still contaminated with HeLa, which definitely changes the proteome and transcriptome of the liver cells present in this cell line.

As a reviewer, I insist: for the sake of reliability of this article, information about the contamination of Chang's liver cell line MUST be included and highlighted as a significant as a significant limitation of this study. The paragraph "In vitro Western blot analysis of Chang liver cells..." is not a satisfactory explanation and I suggest to remove it. And I say this for the last time - I won't accept this article without appropriate corrections.

Please, carefully read the articles cited above (especially the second one), add the  appropriate information in the manuscript, and never use the Chang liver cell line as "normal cells" in the future. It's just wrong.

Author Response

Dear reviewer, the authors of the article apologize to you for their inattention and the unpleasant sensations caused when working with our manuscript. We thank you for your valuable advice and clarification of the situation with Chang liver cells. We will never use them as normal cells for comparison in the future.

The paragraph "In vitro Western blot analysis of Chang liver cells..." was removed.

Moreover, to avoid misunderstandings with Chang liver cells, we conducted additional experiments for the most active substances on a normal WI38 cell culture (diploid human cell strain composed of fibroblasts derived from healthy lung tissue) and added the corresponding data to the “Results and discussion” section. We hope that we have made a better choice in favor of this cell culture. All recent changes are highlighted in blue.

Once again, thank you for your consideration and patience.
